# Identifying Drivers of Predictive Aleatoric Uncertainty

## Abstract

Explainability and uncertainty quantification are two pillars of trustable artificial intelligence. However, the reasoning behind uncertainty estimates is generally left unexplained. Identifying the drivers of uncertainty complements explanations of point predictions in recognizing model limitations and enhances trust in decisions and their communication. So far, explanations of uncertainties have been rarely studied. The few exceptions rely on Bayesian neural networks or technically intricate approaches, such as auxiliary generative models, thereby hindering their broad adoption. We propose a straightforward approach to explain predictive aleatoric uncertainties. We estimate uncertainty in regression as predictive variance by adapting a neural network with a Gaussian output distribution. Subsequently, we apply out-of-the-box explainers to the model's variance output. This approach can explain uncertainty influences more reliably than more complex published approaches, which we demonstrate in a synthetic setting with a known data-generating process. We further adapt multiple metrics from conventional XAI research to uncertainty explanations. We quantify our findings with a nuanced benchmark analysis that includes real-world datasets. Finally, we apply our approach to an age regression model and discover reasonable drivers of uncertainty. Overall, the proposed method explains uncertainty estimates with little modifications to the model architecture and decisively outperforms more intricate methods.

## 1 Introduction

Researchers have recognized the importance of uncertainty quantification and explainability of machine learning (ML) predictions to ensure the successful adoption of ML-based systems in safety-critical applications (Abdar et al., 2021; Vilone & Longo, 2020). These two dimensions are key indicators of a model's trustworthiness, reliability, and fairness, crucial for its broad acceptance (Lambert et al., 2022; Lötsch et al., 2022). Predictive uncertainty in ML refers to the degree of confidence associated with a model's predictions (Chua et al., 2023). It can be decomposed into an epistemic and aleatoric component (Kendall & Gal, 2017). Epistemic uncertainty arises from the scarcity of data in specific areas of the input space, such as the underrepresentation of a particular condition. Principally, it can be reduced by acquiring additional examples. Aleatoric uncertainty refers to inherent randomness or variability in the data, representing uncertainty that cannot be reduced by including additional training instances. It can arise due to measurement errors or certain variables relevant to the observed process not being collected. Uncertainty estimation is critical in risk management. It allows taking conservative action, relying on the model only when it exhibits a high degree of confidence in its predictions, and avoiding usage outside its area of competence (Kompa et al., 2021).

Explainability encompasses methods that enhance the transparency of ML models by highlighting how features influence model output or by rendering the internal computations of black-box models more interpretable. Explainability methods enable understanding whether a model has learned relevant patterns from the input data and can reveal interesting, previously unknown associations (Samek et al., 2021; Schwalbe & Finzel, 2023). Uncertainty quantification and explainability ensure accountable, informed, and responsible decision-making and help mitigate biases and risks (Bhatt et al., 2021; McGrath et al., 2023).

In most applications, explainability focuses on interpreting point predictions (Vilone & Longo, 2020). There is a significant gap in understanding and explaining the drivers of uncertainty estimates. When an ML algorithm is deployed and yields a substantial uncertainty estimate for a specific instance, the possible courses of action involve abstaining from employing the model if alternatives are available or accepting the increased risk. With explainable uncertainties, users gain the capability to identify the factors contributing to elevated uncertainty levels. This understanding allows domain experts to judge their relevance in a given scenario. Additionally, it provides valuable insights into modifications required to augment the model's predictive certainty and performance. In cases where abstaining from model usage is still necessary, factors influencing the decision can be understood and communicated. For example, if such an uncertainty factor is a feature indicating a person's minority status, it could suggest a potential bias of the model. The bias would be undetectable by naive explanations if this feature only influences the uncertainty but not the mean prediction. While detecting and explaining distribution shifts and epistemic uncertainty is an equally interesting problem (Brown & Talbert, 2022), we focus our work on aleatoric uncertainty. Aleatoric uncertainty estimates and explanations are relevant for domains where the noise of the outcome of interest is not constant across independent variables, i.e., heteroscedastic settings. In these cases, aleatoric uncertainty explanations offer complementary information to explanations of point predictions, as the relevant variables influencing mean and variance might differ significantly. Heteroscedastic settings emerge, for example, in the estimation of biophysical variables (Lázaro-Gredilla et al., 2014), the estimation of cosmological redshifts (Almosallam et al., 2016), and robotics and vehicle control (Bauza & Rodriguez, 2017; Smith et al.; Liu et al., 2021).

Explanations can be categorized as either local or global (Schwalbe & Finzel, 2023; Adadi & Berrada, 2018). Local explanations characterize how a model makes predictions for a specific instance. A local explanation of the model's uncertainty could foster more transparent discussions about ML-assisted decisions and risks, increasing trust. Global explanations provide an overview of a model's behavior across the entire input space. They serve to detect general drivers of uncertainty and certainty. These can then be leveraged to formulate hypotheses to improve the model or to detect unintended shortcuts in the uncertainty estimation process, such as spurious correlations or biases.

There is little prior work on explaining uncertainties, and existing literature mainly focuses on classification tasks and generally relies on Bayesian neural networks (BNNs) or technical intricacies such as auxiliary generative models (Antoran et al., 2021; Perez et al., 2022; Ley et al., 2022; Wang et al., 2023). BNNs assign probability distributions to network weights to capture uncertainty (MacKay, 1992). However, due to their computational complexity and involved training process, BNNs have not been as widely adopted as classical neural networks (Lakshminarayanan et al., 2017).

We propose a straightforward and scalable approach for explaining uncertainties in a heteroscedastic regression setting that can be readily integrated into ML pipelines (see Figure 1). We extend point prediction models to additionally estimate parameters of the spread of a given probability distribution. Specifically, we predict parameters of a Gaussian distribution as in a heteroscedastic regression model (Bishop, 1994). The variance parameter of the Gaussian can be interpreted as a measure of the aleatoric uncertainty of the model. We can then use any state-of-the-art explainability method to explain the variance estimate provided by this distributional model. By highlighting input features contributing to the variance output, we identify the inputs contributing to model uncertainty.

Currently, there is a gap in the comparative evaluation of uncertainty explainers in the literature. Therefore, we devise a benchmark with synthetic data with a known data-generating process to analyze a method's ability to detect uncertainty drivers. Further, we extend unsupervised XAI metrics to evaluate uncertainty explainers and perform a second benchmark across diverse datasets. We compare our approach to Counterfactual Latent Uncertainty Explanations (CLUE) (Antoran et al., 2021) and InfoSHAP (Watson et al., 2023). Finally, we apply our approach to an age regression task on images. In summary, our contribution is as follows: We propose a straightforward explanation method for uncertainty and evaluate it against existing approaches. We introduce a benchmark using synthetic data, real data, and synthetically augmented data, including established metrics from the XAI field. Thereby, we provide a resource for informed usage of uncertainty explanation methods.

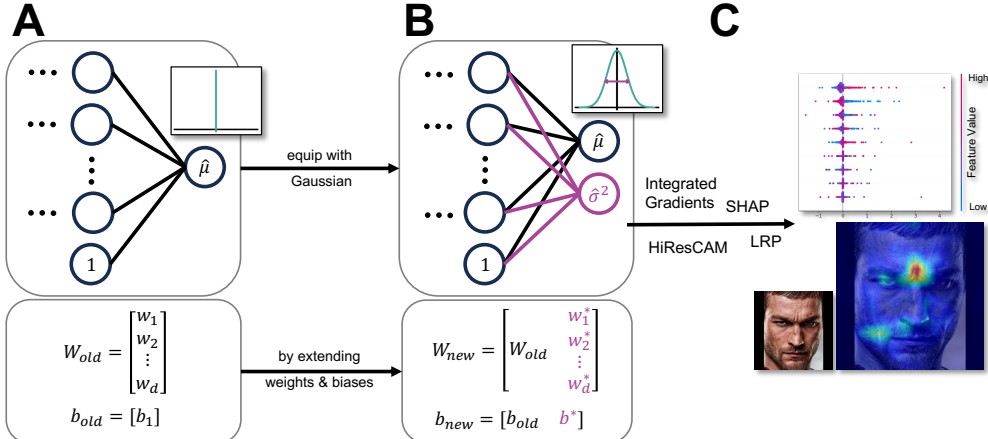

Figure 1: Overview of the variance feature attribution pipeline. (A) A point prediction model with an output layer with weight matrix $\boldsymbol{W}_{old} \in \mathbb{R}^{d \times 1}$ and a scalar bias. We equip this model with a Gaussian distribution resulting in (B), a model with output weight matrix $\boldsymbol{W}_{new} \in \mathbb{R}^{d \times 2}$ and bias $\boldsymbol{b}_{new} \in \mathbb{R}^2$. The two outputs are the mean $\hat{\mu}$ and the variance $\hat{\sigma}^2$ of the predictive distribution. (C) From there, we can explain the variance using any suitable explainability method, resulting in attributions to the input features that can be used to understand the drivers of the model's aleatoric uncertainty.

## 1.1 RELATED WORK

In some research communities, such as causal inference, graphical models, and Gaussian processes, explicitly modeling uncertainty is a prominent area of interest. Furthermore, uncertainty quantification and explainability are rich areas of research within the deep learning field (Abdar et al., 2021; Vilone & Longo, 2020). Yet, few researchers have recognized the importance of explaining the sources of uncertainty in deep learning predictions. Yang & Li (2023) have developed an explainable uncertainty quantification approach for predicting molecular properties. They employ message-passing neural networks and generate unique uncertainty distributions for each atom of a molecule. This approach is inherently specialized for graph-based representations of molecules. CLUE (Antoran et al., 2021) and related approaches (Perez et al., 2022; Ley et al., 2022) derive counterfactual explanations by optimizing for an adversarial input that is close to the original input but minimizes uncertainty. The adversarial input is constrained to the data manifold with a deep generative model of the input data to prevent out-of-distribution explanations. This requires an optimization process for each explanation and the training of an auxiliary generative model, rendering CLUE and its extensions computationally demanding and difficult to implement. Additionally, Antoran et al. (2021) developed an evaluation method for contrastive explanations of uncertainty. Wang et al. (2023) have developed a gradient-based uncertainty attribution method for image classification with BNNs. They modify the backpropagation to attain complete, non-negative pixel attribution and to prevent vanishing gradient issues. To detect and explain model deterioration, Mougan & Nielsen (2023) use classical ML methods and bootstrapping. They train a model and obtain uncertainty estimates on a test set transformed with an artificial distribution shift. In a second step, they train another model to predict the uncertainty estimates from the first step. Subsequently, Shapley values are estimated for the second model to explain the uncertainty. Mehdiyev et al. (2023) employ quantile regression forests to obtain prediction intervals that quantify uncertainty. They extract feature attributions for the uncertainty by estimating Shapley values directly for these prediction intervals as output. Watson et al. (2023) introduce variants of the Shapley value algorithm to explain higher moments of the predictive distribution by quantifying feature contributions to conditional entropy. They use a split conformal inference strategy. They first train a base XGBoost model to predict conditional probabilities. Subsequently, they fit an auxiliary model to the base model residuals. They interpret the estimated Shapley values of this residual model as uncertainty explanations.

## 2 METHODS

### 2.1 DEEP HETEROSCEDASTIC REGRESSION AND EXTENSION OF PRE-TRAINED MODELS

We want to explain uncertainties in neural network regressors. For this purpose, we leverage their extension to deep distributional neural networks, which estimate the parameters of a specified output distribution. For this work, we focus on deep heteroscedastic regression with a Gaussian output, where we capture the mean and variance of the target, thereby directly modeling input dependence of the output noise. Here, we consider a regression setting with $n$ independent training examples $\{(\boldsymbol{x}_i, y_i)\}_{i=1}^{n}$ with input feature vector $\boldsymbol{x}_i \in \mathbb{R}^k$ and target $y_i \in \mathbb{R}$, $i = 1, \ldots, n$. Instead of providing a complete picture of the conditional distribution of the target, deep regression models usually only estimate its conditional mean by optimizing the mean squared error (MSE) or comparable loss functions. In contrast, we assume a heteroscedastic Gaussian as the conditional distribution $y \mid \boldsymbol{x} \sim \mathcal{N}\left(\mu_{\boldsymbol{x}}, \sigma_{\boldsymbol{x}}^2\right)$ and represent its mean $\mu_{\boldsymbol{x}}$ and variance $\sigma_{\boldsymbol{x}}^2$ using a neural network $f_{\boldsymbol{\theta}} : \mathbb{R}^k \to \mathbb{R} \times \mathbb{R}^+$ with weights $\boldsymbol{\theta}$ and two output neurons producing the mean and variance estimates $f_{\boldsymbol{\theta}}\left(\boldsymbol{x}\right) = (\hat{\mu}_{\boldsymbol{x}}, \hat{\sigma}_{\boldsymbol{x}}^2)$, respectively. As first described by Bishop (1994), we can then optimize the Gaussian negative log-likelihood (GNLL): $\mathcal{L} \propto \sum_{i=1}^{n} \left( \log(\hat{\sigma}_{\boldsymbol{x}_i}^2) + \frac{(y_i - \hat{\mu}_{\boldsymbol{x}_i})^2}{\hat{\sigma}_{\boldsymbol{x}_i}^2} \right)$ and interpret the predicted variance as a measure of the aleatoric uncertainty of the model. However, naively optimizing this criterion with overparametrized models such as deep neural networks can be unstable. These models frequently overfit by excessively shrinking the variance estimate or underfit by predicting only a mean estimate of the target and fitting the variance to the overall target variation in the data (Kuprashevich & Tolstykh, 2023; Wong-Toi et al., 2023; Nix & Weigend, 1994; Seitzer et al., 2022). In practice, these convergence difficulties can be mitigated by initially training the model using solely the MSE $\sum_{i=1}^{n} \left( y_i - \hat{\mu}_{\boldsymbol{x}_i} \right)^2$ and subsequently switching to the GNLL (Sluijterman et al., 2023).

The two-stage training process aligns with transfer learning: MSE-based initial training serves as pre-training, followed by fine-tuning with the GNLL to capture predictive uncertainty. Extending existing pre-trained models to capture uncertainty is relevant when the model size and associated training costs make full re-training unfeasible. Pre-trained regression models can be extended by concatenating a column of randomly initialized weights to the weight matrix of the output layer to attain a variance estimate (see Figure 1).

### 2.2 POST-HOC EXPLANATION OF PREDICTIVE VARIANCE

Classic explainability methods explain the predicted class or point prediction. In contrast, we want to explain the variance output in a heteroscedastic regression model. In these models, variance is an additional output to which we can apply any existing, appropriate explainability method. In principle, an uncertainty explanation can be achieved for any parametrized output distribution for which an explicit formulation of the uncertainty is available. In the case of a Gaussian output distribution, the application is most intuitive since the variance, as a parameter of the Gaussian, is a direct output of the neural network. Furthermore, unlike distributions such as the Poisson or exponential distributions, the variance is uncoupled from the mean output.

We employ model-agnostic and model-specific post-hoc explainability methods to explain uncertainty. Model-specific methods are limited in the type of models that they can explain but may offer advantages such as lower computational complexity. In contrast, model-agnostic methods can be applied to any model (Adadi & Berrada, 2018). For our experiments, we combine the approach described in Section 2.1 with multiple explainability methods and refer to this conjunction as Variance Feature Attribution (VFA) flavors. As the first explainability method, we use KernelSHAP (Lundberg & Lee, 2017), a model-agnostic, local explainability method. KernelSHAP approximates Shapley values using a weighted linear surrogate model with an appropriate weighting kernel (VFA-SHAP). We also employ Integrated Gradients (IG) (Sundararajan et al., 2017), which is a local, model-specific method and assigns feature importance by integrating predictions over a straight path from a baseline to the input (VFA-IG). Further, we use Layer-Wise Relevance Propagation (LRP) (Bach et al., 2015), a local, model-specific explainability method developed for neural networks where the importance is distributed backward to the input layer by layer weighted by a neuron's contribution (VFA-LRP). We compare the VFA flavors to CLUE, for which we have to train a variational autoencoder on the train data and apply the optimization as detailed by Antoran

et al. (2021). We calculate CLUE feature attributions as the absolute differences between the feature vector of the counterfactual and its original input feature vector. CLUE is local and model-specific. Further, we reimplement InfoSHAP for regression, which estimates the uncertainty of an XGBoost model by a second auxiliary XGBoost model trained on the residuals of the base model. The uncertainty attribution is attained by estimating the Shapley values of the auxiliary model (Watson et al., 2023). As InfoSHAP builds on SHAP, it is a model-agnostic, local explainability method. We can also attain global explanations from these local methods by averaging explanations for each feature over a given dataset. For the age detection experiment, we use VFA with HiResCAM (Zhou et al., 2016; Draelos & Carin, 2021). HiResCAM is a local and model-specific method that extracts an explanation by weighting the last feature map with the gradient of the output with respect to this last layer feature map. We follow Chefer et al. (2021) and apply HiResCAM to the $[CLS]$ token of the last attention layer.

## 2.3 UNCERTAINTY EXPLANATION EVALUATION METRICS

There is little prior work on evaluating the quality and properties of uncertainty explanations. Generally, high-quality explanations have to be robust, faithful, and highlight relevant input features. We extend established metrics for general XAI to the explanation of model uncertainty (Alvarez-Melis & Jaakkola, 2018; Arras et al., 2022).

In a situation where ground truth noise drivers are known, we can examine if explanation methods correctly rediscover them. Arras et al. (2022) introduce metrics for this setting for classical XAI: *Relevance Rank Accuracy* describes the number of known relevant features that are discovered by the explanation method for a given sample $\boldsymbol{x}_i$. It is defined as

$$\text{RRA}(\boldsymbol{x}_i) = \frac{|P_{topK} \cap P_{\text{GT}}|}{|P_{\text{GT}}|}, \tag{1}$$

where $P_{\text{GT}}$ is the set of ground truth relevance features for $\boldsymbol{x}_i$ and $P_{topK} = \{p_1, ..., p_K\}$ is the set of top $K = |P_{\text{GT}}|$ relevant features identified by the explainability method ordered by their relevance $r$. *Relevance Mass Accuracy* describes the total amount of relevance that is assigned to the ground truth features, normalized by the total amount of relevance:

$$\text{RMA}(\boldsymbol{x}_i) = \frac{\sum_{p_k \in P_{\text{GT}}} r_{p_k}}{\sum_{k=1}^{M} r_{p_k}}, \tag{2}$$

where $M$ is the number of features in the dataset and $r_{p_k}$ the relevance of a feature $p_k$ for a sample $\boldsymbol{x}_i$. For uncertainty explanations, we judge if a method discovers features that correlate with the standard deviation of the target's heteroscedastic noise. To scrutinize global explanations, we apply these accuracy metrics to global feature attributions, giving rise to global relevance rank accuracy (GRA) and global relevance mass accuracy (GMA). Global accuracy measures how effectively a model detects general drivers of uncertainty across the entire dataset. In contrast, local accuracy indicates the model's ability to identify uncertainty sources for individual instances and is, therefore, a stricter criterion.

Alvarez-Melis & Jaakkola (2018) argue that *Robustness* is a key property of explanations, demanding that proximal inputs lead to similar explanations. They propose to evaluate robustness with local Lipschitz continuity:

$$\hat{L}(\boldsymbol{x}_i) = \max_{\boldsymbol{x}_j \in \mathcal{N}_\epsilon(\boldsymbol{x}_i)} \frac{\|f(\boldsymbol{x}_i) - f(\boldsymbol{x}_j)\|_2}{\|\boldsymbol{x}_i - \boldsymbol{x}_j\|_2}, \tag{3}$$

where $f$ is the explanation method. For a dataset with only continuous features, the perturbation space $\mathcal{N}_\epsilon(\boldsymbol{x}_i)$ is a ball with radius $\epsilon$ around sample $\boldsymbol{x}_i$. However, continuous perturbations lack meaning for categorical features. Instead, the perturbation space is defined as the set of data points close to $\boldsymbol{x}_i$: $\mathcal{N}_\epsilon(\boldsymbol{x}_i) = \{\boldsymbol{x}_j \in \mathcal{X} \mid \|\boldsymbol{x}_i - \boldsymbol{x}_j\| \leq \epsilon, \boldsymbol{x}_i \neq \boldsymbol{x}_j\}$, where $\mathcal{X}$ is the set of test inputs. Low Lipschitz estimates indicate small changes in the explanation upon perturbation and, therefore, high robustness. This notion of robustness can be extended to explanations of uncertainty by applying it to the variance head predictions or an auxiliary uncertainty model.

Further, we analyze *Faithfulness* of the explanations. If an explanation is faithful, changing input features that are considered relevant should lead to a significant reduction in prediction performance.

Commonly, this is measured as the increase of the loss upon perturbation of relevant features (Arras et al., 2022). However, the GNLL loss we use during training is a function of the mean and variance and its magnitude is not interpretable. We aim to evaluate the perturbation's impact on the quality of the uncertainty estimate. Naturally, we demand that a higher uncertainty estimate should relate to a higher expected squared error of the mean prediction. Therefore, we measure the correlation between the squared residuals and the uncertainty estimates. Precisely, we calculate the Spearman correlation $\rho_s = \text{corr}_s\left((\boldsymbol{y} - \hat{\boldsymbol{\mu}}(\boldsymbol{X}))^2, \hat{\boldsymbol{\sigma}}^2(\boldsymbol{X})\right)$. We determine the globally most important uncertainty features as the features with the highest feature attribution when averaged over all test instances. For each instance, we add noise to these uncertainty features to attain a perturbed design matrix $\boldsymbol{X}'$. We calculate the correlation of the original residuals with the perturbed uncertainties $\rho_s' = \text{corr}_s\left((\boldsymbol{y} - \hat{\boldsymbol{\mu}}(\boldsymbol{X}))^2, \hat{\boldsymbol{\sigma}}^2(\boldsymbol{X}')\right)$ and expect the variance to be less expressive after the perturbation, i.e., the change $\rho_s' - \rho_s$ is negative.

## 2.4 BENCHMARKING DATA: SYNTHETIC DATA GENERATION AND REAL-WORLD DATASETS

### 2.4.1 SYNTHETIC DATA GENERATION

Evaluating explainability methods on real-world data is challenging due to the subjective nature of interpreting explanations based on expert prior knowledge. To address this, we employ synthetic data with a known data-generating process. Thereby, we can introduce controlled sources of heteroscedasticity, which we aim to detect. Specifically, we sample a synthetic ground truth using a linear system $\boldsymbol{\mu} = \boldsymbol{V}\boldsymbol{\beta}$ with a design matrix $\boldsymbol{V} \in \mathbb{R}^{n \times p}$ with $\boldsymbol{V}_{ij} \sim \mathcal{N}(0, 1)$, and ground truth coefficients $\boldsymbol{\beta} \in \mathbb{R}^p$ with $\boldsymbol{\beta}_i \overset{\text{iid}}{\sim} \text{Uniform}([-1, 1])$. We introduce heteroscedastic noise sources with an absolute-value transformed polynomial model for the heteroscedastic noise standard deviation: $\boldsymbol{\sigma} = |\phi(\boldsymbol{U})\boldsymbol{\gamma} + \boldsymbol{\delta}|$, whereby $\boldsymbol{U} \in \mathbb{R}^{n \times p'}$ is a design matrix with $\boldsymbol{U}_{ij} \overset{\text{iid}}{\sim} \mathcal{N}(0, 1)$,

$$\phi(u_1, u_2, \ldots, u_{p'}) \rightarrow (1, u_1, \ldots, u_{p'}, u_1^2, u_1 u_2, \ldots, u_{p'}^2)$$

is a second degree polynomial feature map, and $\boldsymbol{\delta} \sim \mathcal{N}\left(\boldsymbol{0}, \sigma_\delta^2 \boldsymbol{I}\right)$ is the uncertainty model error. $\boldsymbol{\gamma} \in \mathbb{R}^{\binom{p'+2}{2}}$ are ground truth noise coefficients with entries sampled from $\boldsymbol{\gamma}_i \sim \text{Uniform}([-1, -0.5] \cup [0.5, 1])$ to avoid negligible effects. We can then sample the target $\boldsymbol{y} \in \mathbb{R}^n$ with

$$\boldsymbol{y} \sim \mathcal{N}\left(\boldsymbol{\mu}, \alpha \cdot \text{diag}(\boldsymbol{\sigma}^2) + \sigma_\epsilon^2 \boldsymbol{I}\right). \tag{4}$$

$\alpha \in \mathbb{R}^+$ determines the overall strength of the heteroscedastic uncertainty and $\sigma_\epsilon^2 \in \mathbb{R}^n$ regulates the homoscedastic noise.

For our experiments, we set $\alpha = 2.0$, $\sigma_\epsilon^2 = 0.02$, and $\sigma_\delta^2 = 0.05$ to get non-negligible, feature-dependent noise. We choose $p = 70$ and $p' = 5$ so that the uncertainty sources have to be detected among a larger set of features that do not influence the uncertainty. We sample $n = 41,500$ data points and concatenate both design matrices to attain the input $\boldsymbol{X}_{(n \times 75)} = \left[\boldsymbol{U}_{(n \times 5)}, \boldsymbol{V}_{(n \times 70)}\right]$ which we split into 32,000 train, 8,000 validation, and 1,500 test instances.

In reality, we expect noise features to overlap with features influencing the mean. We separate these in the synthetic data to allow for unambiguous assessments in the evaluation. We included an analysis of a mixed setting in Appendix A.2, where a subset of features simultaneously influences both the mean and variance.

### 2.4.2 REAL WORLD DATASETS

In addition to our synthetically generated data, we incorporate three standard regression benchmark datasets into our evaluation: UCI Wine Quality (Cortez et al., 2009), Ailerons (Torgo, 1999), and LSAT academic performance (Wightman, 1998). These datasets were selected to vary in size and complexity, allowing a comprehensive analysis. We only use data on red wines from the Wine Quality dataset, comprising eleven continuous features for 1,599 samples. The Ailerons dataset has 40 continuous features and 13,750 samples. Encompassing 21,790 samples, the LSAT dataset is the largest dataset considered. It has two continuous features and two one-hot encoded categorical features. We partition all datasets into 70% for training, 10% for validation, and 20% for testing.

## 2.5 BENCHMARKING SETUP

We divide our benchmark into two stages. First, we qualitatively and quantitatively evaluate the uncertainty explanation methods in a controlled setting on a synthetically generated dataset. Second, we investigate the same methods concerning their local RRA and RMA, faithfulness, and robustness on synthetic and real-world data.

In the first stage of our benchmark, we aim to detect global drivers of uncertainty in a synthetic setting. We fit a deep neural network of four hidden layers with 64, 64, 64, and 32 units. The network has two output neurons for the mean and variance prediction. During training, we apply dropout regularization with a dropout probability of 0.1 to the first two layers. We use the Adam optimizer and a batch size of 64. We pre-train using the MSE and fine-tune the model using the GNLL as the loss function. In both training phases, we stop training when improvement on the validation set ceases and choose the model weights that resulted in the smallest validation loss. For each feature, we attain a global feature importance measure as the mean absolute estimated variance feature attributions over all or a specific subset of test instances, which we then analyze using GRA and GMA.

We follow the same model training procedure for the second benchmarking stage, evaluating accuracy, faithfulness, and robustness. Estimating the local RRA and RMA for a given method requires prior knowledge of features affecting the explained quantity, i.e., uncertainty. As this is not the case for our selected real-world datasets, we augment them with synthetic noise that we aim to detect, effectively creating a semi-synthetic setting. For the three real-world datasets, we consider two scenarios. We add five noise features to the datasets and heteroscedastic Gaussian noise to the targets with a standard deviation correlating with the features. Since the real-world datasets are small, we first use a simple noise model where the absolute sum of the noise features is the standard deviation of the noise distribution, a setting referred to as 1-S. In a second scenario, 50-C, we use the more complex polynomial noise model described in Section 2.4.1. To provide more data to the model in the complex noise scenario, we replicate each data point in the train sets 50 times before sampling additional uncertainty features and target noise. For the synthetic datasets, we similarly perform experiments with a simple (S) and complex (C) noise model but without adjusting the dataset size.

We evaluate the robustness of the uncertainty explanation methods for each dataset by estimating the local Lipschitz continuity for 200 randomly selected data points from the test set. For each selected point $x_i$, we compute a local Lipschitz estimate $\hat{L}(x_i)$ by introducing 100 perturbations. For each feature, we sample a perturbation from a uniform distribution centered at the feature value with a range of 2% of the range of the feature in the train set[1]. This is not applicable to LSAT's categorical features. Instead, we resort to the discrete definition of local Lipschitz continuity. Specifically, we compute $\hat{L}(x_i)$ for 200 data points sampled from the test set such that their neighborhood $\mathcal{N}_\epsilon(x_i)$ with $\epsilon = 0.2$ contains more than five instances.

To evaluate faithfulness, we apply standard Gaussian noise to perturb the three globally most important uncertainty drivers of the test data. We use real-world datasets without added noise features or other augmentations for this evaluation. We omit the LSAT dataset as it mainly contains categorical features for which continuous perturbations lack meaning.

We note that we only add synthetic noise to estimate accuracy metrics, i.e. when calculating RRA and RMA. For all other experiments, we use the real-world datasets as is.

## 2.6 APPLICATION TO AGE DETECTION

Age detection finds application in various areas, from security to retail. We apply MiVOLO (Kuprashevich & Tolstykh, 2023), a state-of-the-art vision transformer that achieves best-of-its-class performance in multiple benchmarks. It was designed to tackle age and gender detection simultaneously to leverage synergies. For simplicity, we use a version of the model that only uses face images as input and omits additional body images. We use a pre-trained version of MiVOLO and, following our procedure introduced in Section 2.1, extend the parameter matrices of the MiVOLO head, auxiliary head, and their respective bias terms. We initialize them using a Gaussian distribution following Glorot & Bengio (2010) and a bias of zero. We fine-tune this model using the IMBD-clean dataset

---

[1]Adapted from https://github.com/viggotw/Robustness-of-Interpretability-Methods

(Lin et al., 2022), using the annotations and pre-processing by Kuprashevich & Tolstykh (2023). We use the GNLL and the Adam optimizer with a learning rate of $1e$-5, a weight decay of $1e$-2, and a batch size of 176, which optimizes GPU utilization on the employed hardware. To detect and visualize the drivers of uncertainty in the images, we use HiResCAM as described in Section 2.2.

The code for all experiments is available online[2] and we provide details on the computational resources used in Appendix A.7.

## 3 RESULTS

### 3.1 BENCHMARKING THE DETECTION OF UNCERTAINTY DRIVERS USING SYNTHETIC DATASETS

We first examine the capability of VFA-SHAP to identify the drivers of uncertainty, which are features that correlate with the magnitude of the heteroscedastic noise. We know the data-generating process for the synthetic dataset and, therefore, the ground truth noise sources. Using this dataset, we find that VFA-SHAP can accurately detect the five ground-truth noise features as primary drivers of uncertainty and that these drivers are distinct from the features most strongly influencing the mean output (Figure 2 A and B).

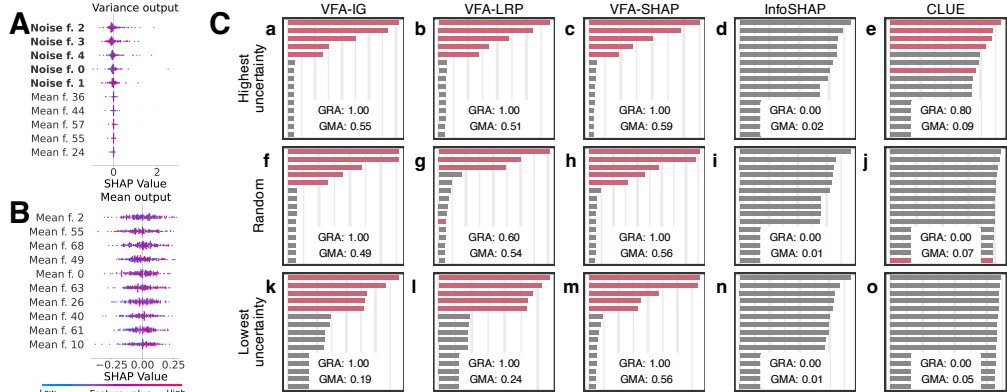

Figure 2: Identifying sources of uncertainty on a synthetic dataset with 70 mean and five noise features. A, B: Explanations of the top 10 most important features (sorted by average absolute estimated Shapley values) for 200 random instances for uncertainty (A) and mean (B) predictions using VFA-SHAP. VFA-SHAP identifies all noise features driving the model's aleatoric uncertainty. Explaining the mean output offers complementary information but disregards noise features. C: Top 15 global importance features with GRA and GMA for each uncertainty explainer. a-e: From the test set of 1,500 samples, we explain the 200 instances with the highest predicted uncertainty. VFA flavors (a, b, c) and CLUE (e) accurately assign high importance to the ground truth noise features (red). f-j: For 200 random instances, VFA delivers close to accurate explanations, while CLUE is unreliable. k-o: VFA also can attribute (un-)certainty for the 200 instances with the lowest uncertainty predictions, while CLUE performance deteriorates. InfoSHAP (d, i, n) does not detect any ground truth noise features.

Further, we analyze the global uncertainty explanation abilities of all VFA flavors, CLUE and Info-SHAP (Figure 2 C). CLUE is applied to the same neural network as VFA, whereas InfoSHAP utilizes XGBoost. Uncertainty estimation may facilitate cautious model application only at high certainty or opting out of model usage due to substantial uncertainty. Therefore, in addition to 200 random instances, we apply the explainers to the test set's 200 highest and lowest uncertainty instances. We find that VFA flavors and CLUE effectively identify uncertainty drivers for high-uncertainty instances, reflected by their GRAs ($K = 5$) close to 1. VFA flavors exhibit superior GMA, which signifies their capacity to disregard irrelevant features. VFA performs reliably for random and low uncertainty examples, while CLUE's performance deteriorates. This suggests that,

---

[2] https://anonymous.4open.science/r/drivers-of-predictive-aleatoric-uncertainty-C6C9

unlike CLUE, VFA can explain the factors contributing to certainty. InfoSHAP does not detect globally relevant uncertainty drivers in these experiments. We find similar results for our synthetic dataset that additionally includes mixed features influencing both mean and variance. VFA flavors considerably outperform CLUE and InfoSHAP in all three settings (see Appendix A.2).

## 3.2 LOCAL ACCURACIES, FAITHFULNESS, AND ROBUSTNESS

We evaluate the local RRA and RMA for the real-world datasets and the synthetic dataset in two settings, one simple (1-S, S) and one complex (50-C, C) as described in Section 2.5 (see Table 1). VFA-SHAP consistently outperforms the other explainers over all datasets. Generally, VFA of any flavor performs best, with CLUE only outperforming VFA-IG and VFA-LRP for the small Red Wine dataset. The difference in accuracy is especially notable for the scenarios with synthetic data. A potential reason is this dataset's smaller ratio of relevant features, i.e., five noise features amongst 70 mean features. As shown in Figure 2, InfoSHAP and CLUE assign similar importance to all

Table 1: Average local RRA ($K = 5$) and RMA over all test set instances for all considered uncertainty explainers and datasets (1-S: simple noise model and original train set, 50-C: complex noise model and artificially enlarged train set; see Appendix A.3 for standard error). The best-performing method is bold. VFA flavors outperform InfoSHAP and CLUE in most settings. VFA-SHAP consistently demonstrates the best performance.

|  |  | Red Wine | | Ailerons | | LSAT | | Synthetic | |
|  |  | 1-S | 50-C | 1-S | 50-C | 1-S | 50-C | S | C |
|---|---|---|---|---|---|---|---|---|---|
| **Average local RRA** | VFA-IG | 0.53 | 0.64 | 0.83 | 0.77 | 0.75 | 0.82 | 0.77 | 0.40 |
|  | VFA-LRP | 0.54 | 0.62 | 0.81 | 0.73 | 0.76 | 0.81 | 0.76 | 0.42 |
|  | VFA-SHAP | **0.80** | **0.92** | **0.87** | **0.91** | **0.93** | **0.94** | **0.87** | **0.75** |
|  | InfoSHAP | 0.36 | 0.42 | 0.34 | 0.35 | 0.73 | 0.73 | 0.11 | 0.12 |
|  | CLUE | 0.59 | 0.67 | 0.54 | 0.59 | 0.51 | 0.50 | 0.07 | 0.06 |
| **Average local RMA** | VFA-IG | 0.49 | 0.69 | 0.80 | 0.80 | 0.75 | 0.91 | 0.52 | 0.26 |
|  | VFA-LRP | 0.50 | 0.66 | 0.78 | 0.77 | 0.77 | 0.91 | 0.50 | 0.28 |
|  | VFA-SHAP | **0.78** | **0.93** | **0.88** | **0.91** | **0.95** | **0.97** | **0.77** | **0.50** |
|  | InfoSHAP | 0.36 | 0.40 | 0.30 | 0.30 | 0.73 | 0.73 | 0.08 | 0.09 |
|  | CLUE | 0.53 | 0.62 | 0.40 | 0.48 | 0.49 | 0.52 | 0.07 | 0.07 |

features. They are, therefore, less selective than VFA, potentially causing uncertainty features not to be detected for many instances. It is worth noting that InfoSHAP achieves higher accuracies than CLUE for the LSAT dataset. LSAT is the only dataset containing categorical features, which might benefit InfoSHAP's XGBoost models.

To analyze the robustness, we calculate distributions of local Lipschitz continuity estimates $\hat{L}(\boldsymbol{x}_i)$ over 200 randomly chosen test set instances for each dataset and method (see Figure 3). According to the obtained Lipschitz estimates, the Shapley-value-based methods VFA-SHAP and InfoSHAP, and VFA-IG seem more robust than CLUE and VFA-LRP. The methods' individual ranking differs between datasets, suggesting that the choice of the most robust method is subject to the dataset.

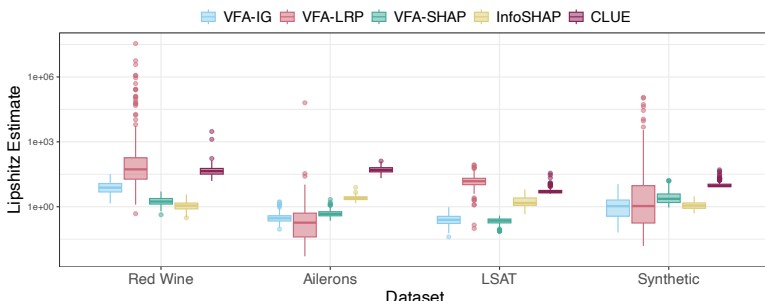

Figure 3: Local Lipschitz continuity estimates for 200 randomly chosen test set instances for all methods and datasets. Lower values indicate higher robustness. VFA-SHAP, InfoSHAP, and VFA-IG are the most robust.

When analyzing the faithfulness metric, we find that perturbation in the most important features faithfully reduces the correlation between uncertainties and residuals (by more than 0.14 in all but one case) when VFA is used in the Ailerons and synthetic datasets (Appendix A.4). This manifests irrespective of the choice of the post-hoc explainer used on the variance output. On these datasets, the explanations of CLUE and InfoSHAP are not faithful (changes in correlations $< 0.01$). On the much smaller Red Wine dataset, InfoSHAP is the most faithful (with a reduction of the correlation of 0.06), while overall, the reductions in correlation are small or zero. In essence, the challenge of learning and explaining uncertainty is amplified in scenarios where data is scarce, leading to suboptimal faithfulness metrics.

### 3.3 EXPLAINING UNCERTAINTY IN AGE DETECTION

Finally, we showcase the application of VFA to age detection using MiVOLO and the IMDB-clean dataset. Applying VFA with HiResCAM reveals reasonable potential explanations for the predictive uncertainty (see Figure 4). The explanations mainly focus on areas around the eyes, mouth, nose, and forehead. These areas are highlighted especially strongly when the person in the image shows emotions that lead to distortions of these facial areas.

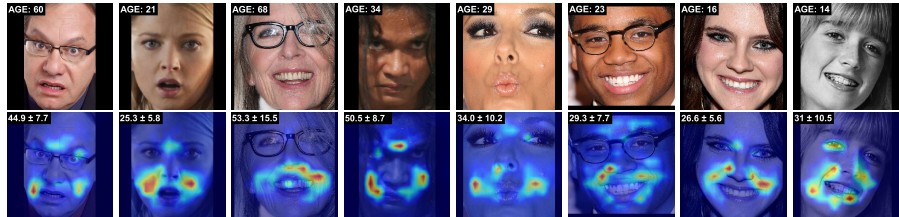

Figure 4: Input images and uncertainty explanations in an age detection experiment using VFA with HiResCAM. Images are annotated with the ground truth and predicted mean and standard deviation.

## 4 DISCUSSION AND LIMITATIONS

We presented a straightforward strategy for explaining predictive aleatoric uncertainties, which requires minimal modifications to existing neural network regressors. We use neural networks with a Gaussian output distribution to estimate uncertainty and apply explanation methods to the variance output. Thereby, we can provide explanations of uncertainty estimates. In synthetic experiments, the resulting explanations outperform alternative methods. As seen in the experiments with low uncertainty instances, we can also explain how features contribute to a model's certainty, which is relevant in high-risk applications. Since conventional evaluation metrics are not always directly applicable, we have introduced an evaluation protocol to assess uncertainty explainers. Parts of our evaluation depend on the knowledge of ground truth noise sources. This necessitated the incorporation of synthetic noise, which may deviate from the arbitrarily complex real-world noise patterns. We extend unsupervised explanation quality metrics for accuracy, faithfulness, and robustness to uncertainty attributions. In our benchmark, VFA compares favorably to CLUE and InfoSHAP. We establish that the selection of the explanation method and the dataset-dependent choice of the uncertainty estimation method are significant variables in generating high-fidelity uncertainty explanations. This is highlighted by the performance of VFA-LRP compared to VFA-SHAP. LRP was initially designed for image data, and a naive application to tabular data might not be ideal. Generally, as we combine deep heteroscedastic regression with existing XAI methods, we inherit all the benefits and limitations of these methods, including computational complexity. Future work might involve the application to non-deep learning models and studying synergies in explaining point and uncertainty predictions. For example, in the context of explainable active learning (Ghai et al., 2021), a visualization of both explainability modes could be beneficial.

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

# A APPENDIX

## A.1 SYNTHETIC DATASET WITH MIXED MEAN AND NOISE FEATURES

We initially separated noise and mean-influencing features in the synthetic data to enable unambiguous evaluations. However, in reality, we expect noise features to overlap with features influencing the mean. Specifically, we add five mixed features to the existing setup. These mixed features are generated using the same polynomial model as described in Section 2.4.1, but they contribute to both the mean and the heteroscedastic noise. Thereby, the dataset comprises 80 features: 70 that exclusively influence the target's mean, five that exclusively affect the noise variance of the conditional target distribution, and five mixed features that impact both aspects.

## A.2 GLOBAL RELEVANCE AND MASS ACCURACY FOR THE SYNTHETIC DATASET WITH MIXED MEAN AND NOISE FEATURES

To understand whether the uncertainty explainers maintain their performance in scenarios with mixed features, we analyze the global relevance rank accuracy (GRA) and global relevance mass accuracy (GMA) of the synthetic dataset where a subset of features influence both the mean and the variance of the target distribution. We expect the uncertainty explainers to detect noise features and mixed features as relevant drivers of uncertainty. Figure 5 depicts the results of this analysis. For this mixed setting, VFA flavors reliably identify features relevant to uncertainty and outperform InfoSHAP and CLUE.

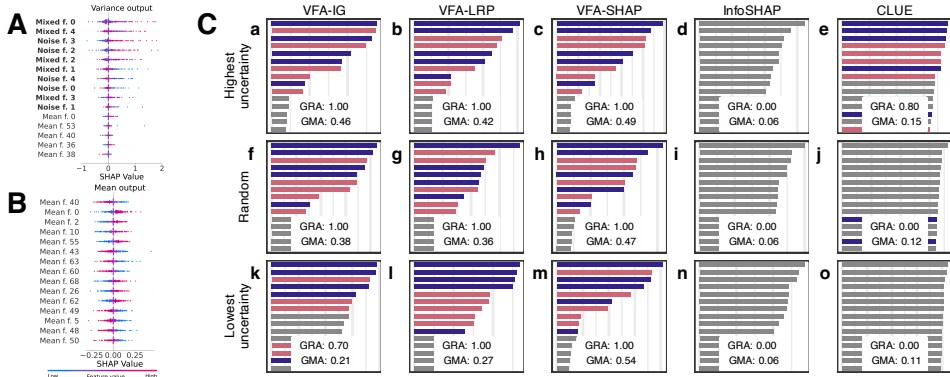

Figure 5: Identifying sources of uncertainty on a synthetic dataset with 70 mean and five noise features and additionally five features that influence the target's mean and variance simultaneously. A, B: Explanations of 200 random instances of the top 10 most important features (sorted by average absolute estimated Shapley values) for uncertainty (A) and mean (B) predictions using VFA-SHAP. VFA-SHAP identifies all noise features driving the model's aleatoric uncertainty. Explaining the mean output offers complementary information but disregards noise features. C: Top 15 global importance features with GRA and GMA for each uncertainty explainer (grey: features that influence the target mean, red: features that influence the noise variance, blue: features that influence the target mean and the noise variance). a-e: From the test set of 1,500 samples, we explain the 200 instances with the highest predicted uncertainty. VFA flavors (a, b, c) and CLUE (e) correctly assign high importance to the ground truth noise features (red) and mixed features (blue). f-j: For 200 random instances, VFA correctly identifies the noise features, while CLUE is unreliable. k-o: VFA also can attribute (un-)certainty for the 200 instances with the lowest uncertainty predictions, while CLUE performance deteriorates. InfoSHAP (d, i, n) does not detect any ground truth noise features.

## A.3 AVERAGE LOCAL RELEVANCE RANK AND RELEVANCE MASS ACCURACY

We calculate the local relevance rank accuracy (RRA) and relevance mass accuracy (RMA) for all datasets for all test samples (see Table 2). We estimate the shown standard error using the `sem()` function of the *scipy* package.

## A.4 FAITHFULNESS OF UNCERTAINTY EXPLANATIONS

Table 3 depicts the faithfulness metric for all datasets with non-categorical features and the uncertainty explanation methods. Perturbation in the most important features faithfully reduces the correlation between uncertainties and residuals when VFA is used in the Ailerons and synthetic datasets. This manifests irrespective of the choice of the post-hoc explainer used on the variance output. On these datasets, the explanations of CLUE and InfoSHAP are not faithful. On the much smaller Red Wine dataset, InfoSHAP performs best, while overall, the reductions in correlation are small or zero. In essence, the challenge of learning and explaining uncertainty is amplified in scenarios where data is scarce, leading to suboptimal faithfulness metrics.

## A.5 ROBUSTNESS OF UNCERTAINTY EXPLANATIONS

We extend our analysis of robustness to the synthetic dataset with mixed mean and noise features. Figure 6 shows that the results for this dataset reflect the observations for the other datasets. The Shapley-value-based methods VFA-SHAP and InfoSHAP, and VFA-IG seem more robust than CLUE and VFA-LRP. The methods' individual ranking differs between datasets, suggesting that the choice of the most robust method is subject to the dataset.

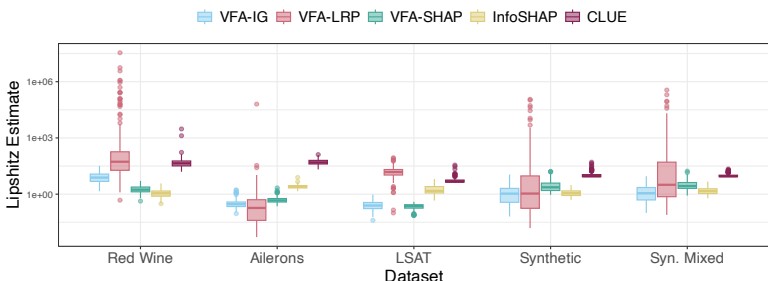

Figure 6: Local Lipschitz continuity estimates for 200 test set instances for all methods and datasets. Lower values indicate higher robustness. VFA-SHAP, InfoSHAP, and VFA-IG are the most robust.

## A.6 UNCERTAINTY QUALITY ANALYSIS

High-quality uncertainty estimates are essential for analyzing explanations of uncertainties. To evaluate the quality of the uncertainty estimates, we first examine the calibration of the uncertainty estimates using the Uncertainty Toolbox[3]. To assess calibration, $\alpha$-prediction intervals are constructed which cover the observed values with a predicted probability of $\alpha\%$. For varying $\alpha$, the calibration plot (Figure 7) shows the predicted proportion of test data within the interval versus the observed proportion of test targets covered by it Chung et al. (2021). We depict the calibration for the synthetic dataset (Figure 7a), the synthetic dataset with mixed uncertainty and mean features (Figure 7b), and for the age detection application (Figure 7c). Overall, the predicted uncertainties are well calibrated but slightly overconfident for the synthetic datasets and marginally too conservative in the age detection task.

For meaningful uncertainty estimates, we also expect that we can reduce the prediction error by restricting the prediction to low-uncertainty instances. Therefore, as depicted in Figure 8, we determine the reduction in root mean squared error on a remaining test set when instances with the highest uncertainty are iteratively removed. We compare this to removal based on the distance of the prediction to the mean prediction as a baseline. For the synthetic datasets, we further depict the deduction in root mean squared error based on the highest ground truth noise standard deviation from the data generating process as the best attainable reduction. We find for the models trained

---

[3]https://github.com/uncertainty-toolbox/

Table 2: Average local RRA ($K = 5$) and RMA ($\pm$ standard error) over all test set instances for all considered uncertainty explainers and datasets (1-S: simple noise model and original train set, 50-C: complex noise model and artificially enlarged train set). %by factor 50. The best-performing method is marked in bold. VFA flavors outperform InfoSHAP and CLUE in most settings, with VFA-SHAP consistently demonstrating the best performance.

| | | Red Wine | | Ailerons | | LSAT | | Synthetic | | Syn. Mixed | |
|---|---|---|---|---|---|---|---|---|---|---|---|
| | | 1-S | 50-C | 1-S | 50-C | 1-S | 50-C | S | C | S | C |
| | VFA-IG | 0.53 ± 0.012 | 0.64 ± 0.011 | 0.83 ± 0.003 | 0.77 ± 0.003 | 0.75 ± 0.003 | 0.82 ± 0.002 | 0.77 ± 0.006 | 0.40 ± 0.006 | 0.59 ± 0.004 | 0.34 ± 0.004 |
| Average | VFA-LRP | 0.54 ± 0.012 | 0.62 ± 0.011 | 0.81 ± 0.003 | 0.73 ± 0.004 | 0.76 ± 0.003 | 0.81 ± 0.002 | 0.76 ± 0.006 | 0.42 ± 0.006 | 0.58 ± 0.004 | 0.34 ± 0.004 |
| local | VFA-SHAP | **0.80** ± 0.007 | **0.92** ± 0.006 | **0.87** ± 0.002 | **0.91** ± 0.002 | **0.93** ± 0.002 | **0.94** ± 0.002 | **0.87** ± 0.004 | **0.75** ± 0.005 | **0.71** ± 0.003 | **0.50** ± 0.003 |
| RRA | InfoSHAP | 0.36 ± 0.010 | 0.42 ± 0.010 | 0.34 ± 0.004 | 0.35 ± 0.004 | 0.73 ± 0.002 | 0.73 ± 0.002 | 0.11 ± 0.004 | 0.12 ± 0.004 | 0.15 ± 0.003 | 0.17 ± 0.003 |
| | CLUE | 0.59 ± 0.010 | 0.67 ± 0.009 | 0.54 ± 0.003 | 0.59 ± 0.003 | 0.51 ± 0.002 | 0.50 ± 0.002 | 0.07 ± 0.003 | 0.06 ± 0.003 | 0.12 ± 0.003 | 0.12 ± 0.003 |
| | VFA-IG | 0.49 ± 0.009 | 0.69 ± 0.010 | 0.80 ± 0.003 | 0.80 ± 0.003 | 0.75 ± 0.003 | 0.91 ± 0.001 | 0.52 ± 0.004 | 0.26 ± 0.004 | 0.50 ± 0.003 | 0.30 ± 0.003 |
| Average | VFA-LRP | 0.50 ± 0.010 | 0.66 ± 0.010 | 0.78 ± 0.003 | 0.77 ± 0.004 | 0.77 ± 0.003 | 0.91 ± 0.001 | 0.50 ± 0.004 | 0.28 ± 0.003 | 0.48 ± 0.003 | 0.30 ± 0.003 |
| local | VFA-SHAP | **0.78** ± 0.006 | **0.93** ± 0.004 | **0.88** ± 0.001 | **0.91** ± 0.001 | **0.95** ± 0.001 | **0.97** ± 0.001 | **0.77** ± 0.003 | **0.50** ± 0.004 | **0.68** ± 0.003 | **0.39** ± 00.3 |
| RMA | InfoSHAP | 0.36 ± 0.006 | 0.40 ± 0.007 | 0.30 ± 0.002 | 0.30 ± 0.002 | 0.73 ± 0.002 | 0.73 ± 0.002 | 0.08 ± 0.001 | 0.09 ± 0.001 | 0.14 ± 0.001 | 0.15 ± 0.001 |
| | CLUE | 0.53 ± 0.008 | 0.62 ± 0.007 | 0.40 ± 0.002 | 0.48 ± 0.002 | 0.49 ± 0.002 | 0.52 ± 0.002 | 0.07 ± 0.001 | 0.07 ± 0.001 | 0.12 ± 0.001 | 0.12 ± 0.001 |

Table 3: Faithfulness of the uncertainty explanations: Change of Spearman correlation between uncertainties and squared residuals when most important features are perturbed. We expect faithful uncertainty explanations to induce a negative change. We exclude LSAT because we define perturbations only for continuous features. Synthetic Mixed is a synthetic dataset where a subset of features influences the mean and the variance of the target distribution.

|          | Red Wine | Ailerons | Synthetic | Synthetic Mixed |
|----------|----------|----------|-----------|-----------------|
| VFA-IG   | 0.00     | -0.15    | -0.20     | -0.16           |
| VFA-LRP  | 0.00     | -0.06    | **-0.22** | -0.15           |
| VFA-SHAP | -0.02    | **-0.17**| -0.16     | **-0.17**       |
| InfoSHAP | **-0.06**| 0.00     | 0.01      | -0.00           |
| CLUE     | 0.00     | 0.00     | -0.01     | -0.00           |

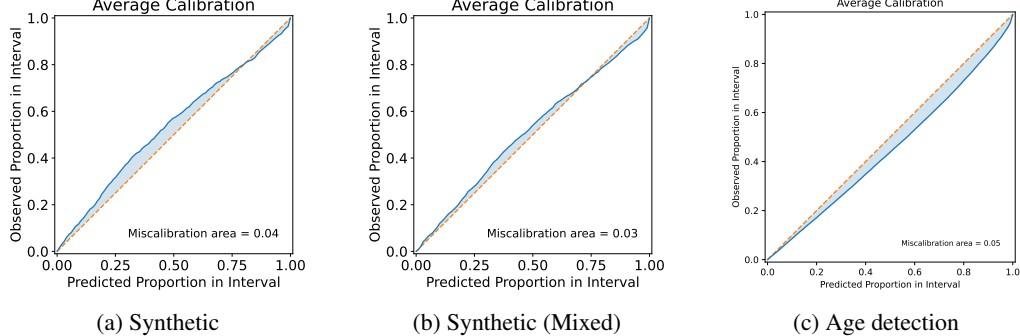

| (a) Synthetic | (b) Synthetic (Mixed) | (c) Age detection |

Figure 7: Calibration plot for models trained on the synthetic and age detection datasets: predicted probability vs. observed proportions of instances covered by various probability intervals. The orange-colored angle bisector marks a perfectly calibrated hypothetical model.

on the synthetic data (Figure 8a), mixed synthetic data (Figure 8b), and the age detection task (Figure 8c) that the uncertainty-based filtering is effective. This indicates that the predicted uncertainty is a meaningful indicator of the expected prediction error.

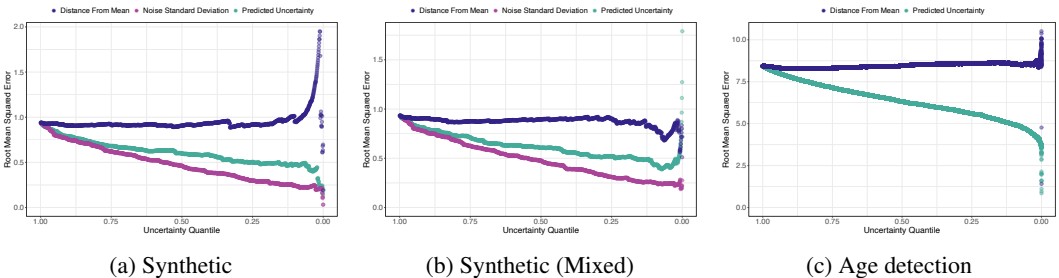

| (a) Synthetic | (b) Synthetic (Mixed) | (c) Age detection |

Figure 8: Root mean squared error of the data for various uncertainty quantiles of the test set. From left to right, data points are removed based on predicted uncertainty (green), distance from the mean prediction (blue), or, for synthetic data, the ground truth noise standard deviation of the data-generating process (purple). For models trained on the synthetic and age detection datasets, lower uncertainty leads to reduced RMSE compared to the overall RMSE of the test set. This indicates that the uncertainty is a meaningful indicator of the expected prediction error.

## A.7 COMPUTATIONAL RESOURCES

We run all experiments on an internal high-performance compute cluster using *SLURM*. The cluster has RTX 2080 Ti (11 GB), Nvidia A40 (48 GB), and Nvidia A100 (40 GB) graphic cards with various RAM configurations. We benchmark global accuracy using a single GPU, CPU, and 32GB of RAM. We used a single GPU, 16 CPUs, and 64GB of RAM to benchmark local accuracy and

robustness. We used a single GPU, CPU, and 200GB of RAM for the faithfulness experiments. We used an Nvidia A100, 16 CPUs, and 48GB of RAM to conduct the age regression experiment.

All experiments (single metric on one dataset) have different run times depending on the dataset size and number of features. However, each experiment ran in less than 24 hours, and the estimated total computation is less than 350 hours. The total project compute, including preliminary experiments, is about twice this time.

