# OpenReview forum: "Identifying Drivers of Predictive Aleatoric Uncertainty"
_ICLR.cc/2025/Conference — ICLR 2025 Conference Withdrawn Submission_

### Official Review · Reviewer_KkVo · 2024-10-22

**Soundness:** 2
**Presentation:** 3
**Contribution:** 1
**Rating:** 3
**Confidence:** 3

**Summary:**

The presented work addressed explainable uncertainty. Previous work on this exists, but no comparative analysis has been done. The authors propose a dataset and a set of metrics where they apply RRA and RMA for uncertainty. They do this by inserting heteroscedastic uncertainty on a tabular dataset. The authors show that methods based on fine-tuning a Gaussian NLL and then running explainability on the variance-head (VFA) outperforms two existing methods (CLUE, InfoSHAP). Authors also show some results where they apply VFA to age regression.

**Strengths:**

- The setup of the method (VFA) and experiments are easy to follow. It is fairly clear what the metrics are measuring and that relevance-accuracy, faithfulness and robustness are desirable properties of an explanation.
- Applying various explainability methods for VFA gives a nice overview of advisable ways to do VFA.
- The language is consistently appropriate and has little to no grammatical/spelling errors.
- The results on the tabular data are persuasive and show strongly that VFA-based methods outcompete InfoSHAP and CLUE for this kind of data, with this type of noise on these metrics.

**Weaknesses:**

- The proposed benchmark only concerns regression with tabular data. I suspect the metrics do not extend well to high dimensional problems such as the age regression example. Additionally, the method (VFA) may not extend very well to classification (though formulations exist, they have limitations), which gives a limitation to the scope of the findings. From your results the conclusion of the comparative analysis is limited to regression on tabular data (where faithfulness is specifically even only for continuous features).
Since the main contribution is the comparative analysis / benchmark, the choices for how the dataset is made and how the metrics work are important, but these are not sufficiently elaborated. Specifically, it needs to be very clear why the heteroscedastic noise is introduced in these ways (1-S, 50-C) and not in some other ways, and how that might impact the results. The metrics are sufficiently understandable, but a more thorough explanation of their properties and potential limitations will make them more persuasive. The choice of datasets used should also be further elaborated (why these datasets and not others / more?).
- The results from the Faithfulness analysis are not in the main body of the paper. Particularly because these results conflict with other results, they should be in the main body.
- The results on face-age regression are not particularly informative. The established criteria (Relevance Rank, Faithfulness and Robustness) are not applied, and the explanations are not comparative between methods and not showing a difference between mean-variance explanations. Additionally, since the “reasonableness” of explanation is determined post-hoc, many possible results could be considered reasonable. If the model would’ve identified uncertainty at the glasses/hair we might conclude that it’s finding occlusion. The suggestion that the highlighted areas relate to emotion are unfounded. High uncertainty is shown in the 3rd, 5th and last face, even though they do not show substantial emotion (other than smiling for a picture), while the second face has clear emotion but a relatively low uncertainty. I’d find the findings from 3.3 as they are uninformative. I’d consider removing them, or expanding substantially by creating a comparative analysis similar to what is done with the tabular data.
- While the authors give some reasoning that aleatoric uncertainty explanations are important (Lines 60-65), the reasoning is not particularly strong or supported by some evidence. There is no evidence that this might be useful in a decision support system for domain experts. The example of minority status presenting a bias in the model seems shaky. A common problem where I might expect this is that facial recognition works poorer on black faces (so high uncertainty attribution to African facial features, or all of the skin tone), but this should be due to epistemic uncertainty not aleatoric uncertainty.

## Additional Feedback (minor comments)

- It should be considered a substantial limitation that the prediction of uncertainty only (theoretically) reflects aleatoric uncertainty and would neglect epistemic uncertainty. This means that some factors that may drive uncertainty in a general sense are ignored. I think this is something that should be discussed in the limitations section of the paper.
- On line 74 “Local or Global” explanations are explained by their purpose, not by how they’re created. A simple definition of local=per sample, global=per dataset would help a reader not familiar with this concept.
Section 1.1 reads like a list of papers, but does not read as a narrative. After reading the rest it makes sense that this section introduce CLUE and InfoSHAP, but this is not clear when reading the paragraph first. Structuring 1.1 into a narrative (with paragraphs) and extracting the meaningful parts will make it easier to read.
- A visualizations of the synthetic data in 2.4.1 and even the synthetic noises would greatly improve interpretability, even if simplified in 2 dimensions. This could be put in an appendix.
- The results from Figure 2 are interesting, but quite hard to interpret. It’s not clear what is “good” behavior for the SHAP values. (large spread, or positive values?), and the plots in C are missing x/y axes labels and a legend (red=uncertainty related feature). The corresponding a-o letters with long caption are also difficult to connect. Perhaps C can be explained by rows=high/mid/low uncertainty samples, columns=Explainable Uncertainty-method, and then discussing the results without referring to the individual subplots.
- There’s a typo in bottom right Table 2 in the appendix.
- The definition for epistemic uncertainty (Lines 39-44) is not entirely accurate. This is not the main point of the paper, but might lead to confusion elsewhere. This definition ignores contributions of model misspecification and approximation error, and does not do good justice to covariate shift. The definition for aleatoric could also be more specific, “randomness in the data” could mean many things. A possible definition for aleatoric uncertainty is stochasticity in the true relationship between X and Y.

**Questions:**

How good are the regression predictions of the different methods? I understand this is not the primary goal, but it might be relevant (especially if MSE for VFA-methods is substantially higher than for InfoSHAP/CLUE)

---

### Official Review · Reviewer_2qst · 2024-11-04

**Soundness:** 3
**Presentation:** 3
**Contribution:** 2
**Rating:** 6
**Confidence:** 3

**Summary:**

The authors propose a method for explaining predictive aleatoric uncertainties (uncertainties due to the inherent variability of the training data) in heteroscedastic regression settings (where aleatoric uncertainty is interesting). Their method generates variances as well as point predictions from the regression NN, then explains the variances using existing XAI methods. They compare to CLUE and InfoSHAP, and also provide a benchmark using synthetic data, real data, and synthetically augmented data.

**Strengths:**

The method is mathematically straightforward and simple to grasp. Figure 1 is a nice conceptual overview. The methods and metrics are well explained (and therefore replicable), and the provided formulas and definitions are appropriately conceptual for understanding the method. In general, the paper organization is very good.

The experimental results, especially on the synthetic data where the noise process is known, are strong compared to the other methods. Results figures are also well-presented; I appreciated the other experimental conditions such as simple vs. complex noise model, or mixed features given in the Appendix.

**Weaknesses:**

My main concern is about originality: how to quantify a contribution that is simply combining existing concepts from explainability and uncertainty quantification, albeit in an appealingly simple and well-presented way. Is good presentation, fairly complete experiments, and mathematical appeal sufficient to call this a major contribution? I’ll say yes because of the performance improvement in identifying noise features and the usability/accessibility of the approach.

There may be a few more experiments that would strengthen this contribution--see Questions.

**Questions:**

Are there quantitative metrics or other methods you can compare with for the age detection explainability task? I agree that the explanations focusing on the eyes, mouth, nose, and forehead is promising.

You present your approach as “straightforward and scalable.” Is there an easy benchmark you can do to demonstrate usability (lower compute) against BNNs? Or even an anecdote of the computational improvement (can cite from elsewhere).

Lakshminarayanan et al., 2017 also proposed ensembles as an alternative non-Bayesian approach for uncertainty quantification. Why is your approach (network learned variance parameter) better?

---

### Official Review · Reviewer_C7vt · 2024-11-04

**Soundness:** 2
**Presentation:** 2
**Contribution:** 2
**Rating:** 3
**Confidence:** 4

**Summary:**

The paper combines deep heteroscedastic regression with model explainability methods to identify potential reasons for aleatoric uncertainty. A framework is built, where to estimate uncertainty, neural regression networks with Gaussian output distribution are used. Explainability methods such as Kernel SHAP, Integrated Gradients, and Layer-Wise Relevance Propagation are then used to explain variance. Based on the synthetic and real-world datasets, the methods are compared to InfoSHAP and CLUE based on a set of metrics. The SHAP-based variance feature attribution method consistently outperforms other approaches.

**Strengths:**

The idea of understanding reasons behind the aleatoric uncertainty is interesting. The paper picks a specific controllable case and studies it.
Multiple evaluation metrics are introduced and both synthetic and real-world datasets are considered for evaluation.
In total, the paper compares five different methods, including two baselines.

**Weaknesses:**

1. The paper introduction is built around understanding aleatoric uncertainty; however, it is not clear from the paper what "understanding uncertainty" is. In general, one would expect that understanding uncertainty means identifying the sources of it. For the case of heteroscedastic regression, the explanations are built by identifying features that contribute to output uncertainty. More discussion is needed on why this method is effective and scalable. The worry is that the methods can be misleading in multiple other cases, for example, when the sources of uncertainty are outputs and not inputs.

2. While applying model explanation methods to estimate uncertainty is an interesting idea, the explainability methods were designed for providing reasons for decision making; therefore, depending on the context, these two problems can be different in nature. More discussion is needed on laying a foundation for why the methods are effective for uncertainty estimation.

3. When the number of uncertainty sources is low (e.g., 5 out of 70) and there is no feature correlation, as in the synthetic dataset, the VFA methods identify the uncertain features. However, it is not clear if this approach will work as well in more realistic scenarios, when the number of uncertainty sources is higher and there are correlations between features.

4. For the synthetic dataset, the ground-truth features are known; for non-synthetic cases, the correlation between the squared residuals and the uncertainty estimates is considered. However, by its nature, this is a post-hoc method that can favor some methods over others. Can you please comment on the reliability of the ground-truth computation?

5. Could you please provide errors in the performance estimates, for example by performing cross-validation over different train and test folds.

Minor:

Line 11: “are two pillars” -> "two of the pillars," as there are many pillars of trustworthy AI that are not mentioned, including interpretability, fairness, robustness, reliability, and so on.

Line 14: Understanding reasons for decision making or uncertainty does not necessarily create trust and can create distrust.

**Questions:**

1. Line 286-287: Could you please elaborate on the need for prior knowledge since the metrics were introduced in Section 2.3?

2. For Figure 4, could you please provide the explanation of the prediction as well? The question is how different the decision explanation and the uncertainty explanation maps are.

3. If XGBoost for InfoSHAP was regularized, it might not choose the uncertainty features for Figure 2. Could you please comment on whether the model contains the uncertainty features?

---

### Official Review · Reviewer_TNHY · 2024-11-07

**Soundness:** 3
**Presentation:** 2
**Contribution:** 2
**Rating:** 5
**Confidence:** 3

**Summary:**

The paper proposes a novel method for explaining predictive aleatoric uncertainty by explaining the variance output in a heteroscedastic regression model. Their method identifies which input features contribute to model uncertainty (i.e., the variance output). Further, they propose benchmarks for evaluation of uncertainty explainers. Lastly, they empirically evaluate their method against existing approaches and demonstrate that the proposed method identifies the noise features driving the model's aleatoric uncertainty.

**Strengths:**

- The paper proposes a novel method for explaining predictive aleatoric uncertainty by explaining the variance output in a heteroscedastic regression model.
- The paper empirically demonstrates that the proposed method VFA-SHAP identifies noise features driving the model's aleatoric uncertainty.

**Weaknesses:**

- A limitation of this method is its computational complexity; can the authors elaborate on the computational cost of their method, relative to existing approaches?
- For the real-world dataset evaluation, the authors show the output explanations of VFA. It is hard to understand the performance of the method on this dataset, since the ground-truth explanations are unknown (which I understand is not available) and only the performance of VFA is presented. It would be useful to show the performance of the other baselines on this real-world task to at least evaluate the performance of this method w.r.t. them.

**Questions:**

- Empirical evaluation is done against only two baselines (CLUE, InfoSHAP); are these the best performing existing methods?

---

### Note · Authors · 2024-11-27

**Comment:**

Dear reviewers,
thank you for taking the time to review our submission. We greatly appreciate your detailed feedback. To fully incorporate your suggestions, we will need more time than the current timeline at ICLR allows. Therefore, we plan to revise the manuscript further and resubmit it to another venue.

**Withdrawal Confirmation:**

I have read and agree with the venue's withdrawal policy on behalf of myself and my co-authors.